# Exploring the Use of Abusive Generative AI Models on Civitai

## ABSTRACT

The rise of generative AI is transforming the landscape of digital imagery, and exerting a significant influence on online creative communities. This has led to the emergence of *AI-Generated Content (AIGC) social platforms*, such as Civitai. These distinctive social platforms allow users to build and share their own generative AI models, thereby enhancing the potential for more diverse artistic expression. Designed in the vein of social networks, they also provide artists with the means to showcase their creations (generated from the models), engage in discussions, and obtain feedback, thus nurturing a sense of community. Yet, this openness also raises concerns about the abuse of such platforms, e.g., using models to disseminate deceptive deepfakes or infringe upon copyrights. To explore this, we conduct the first comprehensive empirical study of an AIGC social platform, focusing on its use for generating abusive content. As an exemplar, we construct a comprehensive dataset covering Civitai, the largest available AIGC social platform. Based on this dataset of 87K models and 2M images, we explore the characteristics of content and discuss strategies for moderation to better govern these platforms.

## CCS CONCEPTS

• **Human-centered computing** → **Empirical studies in collaborative and social computing**.

## KEYWORDS

Social Media, Generative AI, Empirical Study

**ACM Reference Format:**
. 2018. Exploring the Use of Abusive Generative AI Models on Civitai. In *Proceedings of ACM Conference (Conference'17)*. ACM, New York, NY, USA, 10 pages. https://doi.org/XXXXXXX.XXXXXXX

## 1 INTRODUCTION

The commodification of AI Generated Content (AIGC) has had a significant impact on online creative communities [4, 12]. For example, the Generative Diffusion Model (GDM) [35] has achieved state-of-the-art outcomes in the realm of image generation, with open-source implementations like Stable Diffusion [33] easily accessible. Their open-source nature further enables fine-tuning and extension of the models.

This has driven the emergence of *AIGC social platforms* such as Civitai, PixAI, and Tensor.art. These are online platforms for sharing models, images and discussing open-source generative AI. They are designed akin to social media services, allowing users to

showcase their creations, participate in discussions, and receive feedback, thereby creating a sense of community. Uniquely, they also allow users to develop and share their own generative AI models. For instance, bespoke models can be developed for generating particular types of images (*e.g.,* containing particular people or artistic styles) and, subsequently, other users can then share the outputs (images) from these models for further social discussion. These unique features have attracted a significant number of creators sharing numerous novel models and artworks, catalyzing new trends in AI content creation [3, 23].

However, the unrestricted proliferation of diverse models represents a double-edged sword: while they can help unleash creativity, they also pose challenges and risks that require careful consideration. Numerous issues concerning the abuse of generative AI have already been reported, including flooding online communities with not-safe-for-work (NSFW) images [38], disseminating deceptive deepfakes [44], and infringing upon copyright [14]. Anecdotally these platforms have often been the origin of the generative AI models that produce the aforementioned abusive content, and also where the abusive content is initially shared [17, 22]. Thus, the proliferation of abusive content from these platforms can exert a broader influence, permeating other social media communities.

As a result, there is an arguable need to somehow moderate the use of these models on such platforms. However, to date, there have been no prior studies that could inform the debate. With this in-mind, we conduct the first large-scale empirical study of an emerging AIGC social platform, focusing on the *Civitai* — the largest social platform for image models [34]. As of November, 2023, it has attracted 10 million unique visitors each month. We compile a dataset comprising all metadata (for both images and models) shared on Civitai until $15^{th}$, December, 2023, containing 87,042 generative models and 2,740,149 AI-generated images. Using a range of techniques, we then label each model and image with information about its themes and the presence of NSFW concepts. We explore the following research questions:

- **RQ1:** As each model can be highly bespoke, what are the key themes the models are designed to generate images for? Further, what are the subsequent themes of the images generated, and do they reflect a prevalence of abusive content?
- **RQ2:** How popular are models that are designed for generating abusive images, and what types of image prompts do users utilize to generate such content?
- **RQ3:** Are users more active in engaging with abusive models and images, as measured by social metrics such as comments and favorites?
- **RQ4:** Do the creators of abusive models and images exhibit distinct positions within the wider social network (*i.e.* centrality), as compared to creators who do not?

We offer the first characterization of the themes of models and images on Civitai and reveal a prevalence of abusive content. Our main findings include:

(1) We find a range of models (and subsequently generated images) each geared towards a particular theme. 16.97% models and

72.05% images contain tags related to NSFW content; 23.54% models and 32.98% images are deepfakes. Moreover, deepfakes in Civitai tend to be associated with NSFW content (*e.g.,* naked deepfakes), with a positive correlation between tags for NSFW content and deepfakes (model: $\phi = 0.17$; image: $\phi = 0.10$). We also find that over half of the deepfake victims are celebrities.

(2) Models that are designed for NSFW content are more popular than non-NSFW models. On average, NSFW models have generated 36.36 images (per model) *vs.* 24.20 for non-NSFW models. However, we also find that non-NSFW models are frequently re-purposed to generate NSFW content, via prompting. 37.05% of the NSFW images are generated by prompting non-NSFW models to contain NSFW concepts. Additionally, we find frequent references to real person names in the textual description of deepfake models. The most common victims are social media celebrities, such as Instagram influencers or OnlyFans stars.

(3) Civitai users are more active in engaging with NSFW models and images, as measured by common social network metrics. Compared with their non-NSFW counterparts, NSFW models and images receive significantly more downloads/views (models: 3.32x; images: 1.18x), favorites (models: 3.22x; images: 1.63x), and financial "tips" (models: 1.92x; images: 1.53x).

(4) Creators sharing abusive models and images are have higher centrality in the social follower network. For example, creators who have shared at least 3 NSFW or deepfake models/images hold higher median centrality like betweenness (models: 2.59x; images: $1.35e^{-06}$ *vs.* 0), in-degrees (models: 1.50x; images: 6.00x), and PageRank (models: 1.003x; images: 1.005x), compared with those who haven't. Therefore, these creators tend to have more follower links, hold bridge positions and befriend more influential users.

## 2 PRIMER ON CIVITAI

As a social platform, Civitai enables users to share their AI models and generated images, as well as receive feedback, comments and even tips from other users. In this section, we introduce the necessary pre-knowledge about Civitai.

***Models and images.*** Civiati hosts diffusion models and AI-generated images, uploaded by creators. Every model/image is associated with a unique ID and a preview web page public to any users. Various social metadata is visible as well, involving tags (assigned by users or Amazon Rekognition [9, 10]), statistics (*e.g.,* number of downloads/views, likes, and rating scores), and text comments by other registered users. Creators can also attach descriptive information to their models and images, *i.e.* textual descriptions of models' usage and images' configurations, resources, and prompts used for generation.

***Users.*** Similar to common social platforms, Civitai users have profile pages displaying their self-reported information and all their models/images. Users can also attach external links to their profile page, as promotion for their accounts on other social platforms (*e.g.,* Instagram and X) or profitable platforms (*e.g.,* Ko-fi and Patreon). Furthermore, users can follow each other and leave rating scores to the profile pages.

## 3 DATA COLLECTION METHODOLOGY

### 3.1 Data Collection

We compile a dataset containing the metadata of all models, images, and creators on Civitai. To accomplish this, we utilize the Civitai REST API[1] and python Selenium WebDriver.

***Model data.*** We collect 87,042 models' metadata. The metadata contains the number of downloads, likes, comments, and rating score (range from 0 to 5), amount of tips, tags, a textual model description, and flags for whether the model is a real-human deepfake or NSFW. Of these models, 8.0% are checkpoint models (base models), 84.4% are LoRA [11, 19] (or LyCORIS [46]) models (fine-tune models), 5.8% are embeddings and 1.8% are other models.

***Image and prompt data.*** We collect 2,740,149 images' metadata. These are images generated from the shared models. The metadata contains the number of consumers' five reactions (cry, laugh, like, dislike, and heart), number of comments, views, amount of tips, tags, flags for whether the image is NSFW, and the model used to generate the image. Note, as Civitai API does not identify deepfake images, we annotate an image as real-human deepfake if it is generated by any one model explicitly reported as real-human deepfake. Importantly, the metadata also includes the text prompts used for 1,534,922 images.

***Creator data.*** We extract 56,779 creators, with 11,632 model creators and 52,546 image creators (7,399 are both model and image creators). We also gather their lists of followers and followees.

### 3.2 Data Augmentation

We augment our data with two types of annotations: ***(i)*** labeling models and images with their content themes (§4); and ***(ii)*** identifying person names as well as occupations, using the models' descriptions (§5). Considering that our study covers a large-scale dataset, we leverage ChatGPT for this, as relevant literature has highlighted its potential in facilitating theme extraction [16, 42] and person name recognition [40].

***Model implementation.*** We use `gpt-3.5-turbo-0125`. We access the model through OpenAI's API with parameter `temperature` set to 0 to make the response focused and deterministic.

***Extracting tags' themes.*** We first extract the top 500 most popular text tags for the models and images, respectively (note, these are tagged by the model/image uploader, other registered users, and Amazon Rekognition [9, 10]). We then utilize ChatGPT to summarize the potential thematic categories that the tags refer to. Our prompts use a "system" message making ChatGPT respond in a desired JSON format and a "user" message in JSON syntax [48]. We begin by using the first prompt to extract general themes from the tags:

`[SYSTEM MESSAGE]`: You are a helpful assistant designed to output JSON within the desired format: [<potential_theme_of_tags>] `[USER MESSAGE]`: {"Prompt": "The followings are 500 most popular tags associated with shared generative models on a AIGC platform. List potential themes of these tags.", "Tags": ["Tag 1", "Tag 2", ...]}.

Next, we ask ChatGPT to annotate each of the tags with the above extracted themes:

---

[1]https://github.com/civitai/civitai/wiki/REST-API-Reference.

**[SYSTEM MESSAGE]**: You are a helpful assistant designed to output JSON within the desired format: {"Tag": <tag_to_categorize>, "Theme": <theme_of_the_tag>} **[USER MESSAGE]**: {"Prompt": "Categorize the following tag based on the given themes.", "Tag": "Tag input"}.

Following this, two authors manually review ChatGPT's responses to correct any mis-classified tags and consolidate duplicate categories (*e.g.,* merging tags in the "Erotic Art" and "Mature Content" categories into a new category named "NSFW content"). In all, ChatGPT extracts 158 themes from all tags (models: 47; images: 111) and we consolidate them into 6 general categories – "Human attributes" (28 themes), "Deepfakes" (5 themes), "NSFW content" (8 themes), "Virtual character, fictional content, entertainment media" (32 themes), "Scenery objects, decoration, clothing" (28 themes), "Style of art and culture" (48 themes).[2]

***Person name recognition.*** To inspect who are the victims targeted by deepfake models, we leverage ChatGPT to extract real people's names from each model's description. For this, we utilize the following prompt:

**[SYSTEM MESSAGE]**: You are a helpful assistant designed to output JSON within the desired format: {"Entities": [{"Name": <personal_named_entity>, "Occupation": <occupation_of_the_person>}]} **[USER MESSAGE]**: {"Prompt": "Identify all real person names with their occupations.", "Text": "Text input"}

To validate the results, we manually label person names from 100 randomly sampled models. We find that ChatGPT reports the correct occupations for *all* the person names. For additional context, we then group these celebrities by their occupations and rank the groups by the number of derived images. We manually review the top-100 groups and consolidate duplicate groups by standardizing their occupation names (*e.g.,* merging all groups containing the word "actor" into a general group named "actor").

### 3.3 Baseline Prompt Datasets

Our study also contains a later comparative analysis of the usage of NSFW content in prompts on Civitai vs. two mainstream AIGC platforms: Stable Diffusion and Midjourney. For this, we make use of two existing prompt datasets: *DiffusionDB* (1,528,512 distinct prompts from Stable Diffusion Discord) [39] and *JourneyDB* (1,466,884 distinct prompts from Midjourney) [36]. Each of the two datasets contains a large volume of user-generated prompts, allowing us to understand the prompts used on those platforms.

For this, we also employ OpenAI's moderation API, configured with the `text-moderation-006` model, to quantify the degree of NSFW content exposed in each prompt's text across our Civitai dataset, plus *DiffusionDB* and *JourneyDB*. OpenAI's moderation API takes a prompt's text as an input and then reports if it is NSFW content (confidence score ranging from 0 to 1), as well as a flag defining whether the prompt finally classified as NSFW. We choose this moderation model because it is effective in detecting NSFW content [25]. Indeed, it is already used to moderate ChatGPT's prompt input [31].

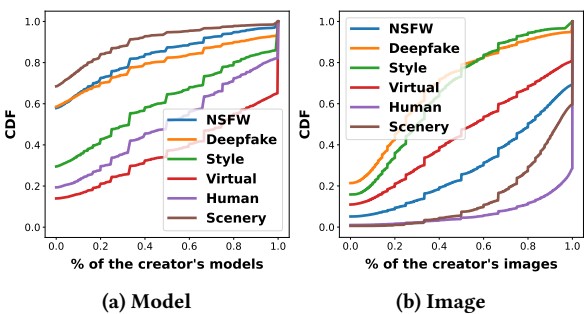

(a) Model                    (b) Image

**Figure 1: CDF of the percentage of models and images, by the creator that contain a specific theme.**

## 4 EXPLORING MODEL AND IMAGE THEMES (RQ1)

In this section, we investigate RQ1, inspecting the themes of models and images. Our ultimate goal is to explore the potential prevalence of abusive content on Civitai.

### 4.1 Overview of Themes

Recall, we use ChatGPT to identify the theme of each model and image, based on their tags. Thus, we begin by examining the distribution of the six identified themes. Table 1 summarizes the themes of models and images extracted by ChatGPT, as well as corresponding statistics. Overall, while "Human attributes" is the most common themes across both models and images, we observe that the focus on themes varies between the creation of models *vs.* images. Model development predominantly revolves around three themes: "Virtual character, fictional content, entertainment media" (65.03% models), "Human attributes" (53.85% models), and "Style of art and culture" (34.58% models). In contrast, image creation focuses on "Scenery objects, decoration, clothing" (87.53% images) and "NSFW" (72.05% images). This suggests that the interests of the model creators differs from the image creators who use those models.

Worryingly, we also note that intuitively abusive themes play a major role in both models and image creation. "Deepfakes" emerge as a significant theme, covering 23.54% of models and 32.98% of generated images. A notable portion of models (16.97%) and images (72.05%) are tagged with the theme "NSFW content". This suggests that creative communities within Civitai may face issues with "abusive" material, indicating a need for enhanced moderation efforts (especially regarding deepfakes and NSFW content).

### 4.2 Overview of Creators' Themes

We next examine the specific themes that each creator (primarily) focuses on. To do this, for each user, we calculate the proportion of each theme within their content portfolio. To reduce noise, we exclude less active creators, who have fewer than 3 models or images.

Figure 1a displays the CDF of the percentage of models by the creator that contain each specific theme (see §3.2 for information

---

[2]We detail the full breakdown in the supplementary material.

| Theme | Model | | | Image | | |
|-------|------|---------|-----------------|------|----------|-----------------|
| | #Tag | #Models | Representatives | #Tag | #Images | Representatives |
| **Human attributes** | 68 (16.80%) | 46,875 (53.85%) | woman, female, girls, man, male | 127 (25.40%) | 2,537,481 (92.91%) | woman, solo, female, person, lips, head, face |
| **Deepfakes** | 35 (7.00%) | 20,490 (23.54%) | celebrity , actress, model, real person, realism | 25 (5.00%) | 900,796 (32.98%) | bands, realistic |
| **NSFW content** | 39 (7.80%) | 14,773 (16.97%) | sexy, nsfw, hentai, porn, pornstar | 61 (12.20%) | 1,967,678 (72.05%) | breasts, sexy attire, adult, nudity, large breast |
| **Virtual character, fictional content, entertainment media** | 175 (35.00%) | 56,602 (65.03%) | character, anime, video game, cartoon, furry | 25 (5.00%) | 1,507,389 (55.20%) | anime, comics, animal ears, cosplay, manga |
| **Scenery objects, decoration, clothing** | 44 (8.80%) | 7,803 (8.96%) | clothing, buildings, vehicle, landscape, architecture | 221 (44.20%) | 2,390,418 (87.53%) | clothing, outdoors, jewelry, cleavage, earrings |
| **Style of art and culture** | 114 (22.80%) | 30,095 (34.58%) | style, lora, base model, portraits, art | 26 (5.20%) | 937,534 (34.33%) | photography, blurry, art, fashion, painting |
| **Miscellaneous** | 25 (5.00%) | 8,025 (9.21%) | concept, tool, fully automated, objects, animals | 15 (3.00%) | 321,586 (11.78%) | animal, bara, mammal, electronics, leather |

Table 1: A summary of thematic categories of models and images on Civitai. The theme "Miscellaneous" is for those tags unable to be coded with any one of the six themes identified by ChatGPT. The column "#Tag" presents the number of tags involved in corresponding themes. "#Models/#Images" presents the number of models/images containing corresponding themes. "Representatives" presents exemplar tags.

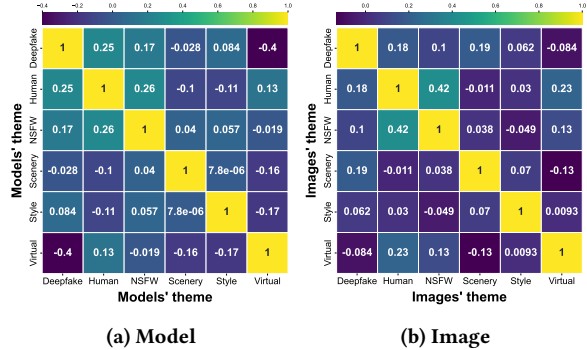

**(a) Model**  **(b) Image**

Figure 2: Heat maps of phi coefficient showing pair-wise category correlation between each of the six themes.

about themes). We observe that the theme "Virtual character, fictional content, entertainment media" is the focus for the majority of model creators, covering a median of 75% of each creator's models. In fact, 34% of creators fully (100%) focus on this theme. In contrast, NSFW and Deepfake models are not the primary focus for most creators, with over 58% of creators not creating any NSFW or Deepfake models. However, the situation for images is very different from models. Figure 1b displays the CDF of the percentage of images by the creator that contain a specific theme. Unlike models, NSFW is a very popular theme for image creators: 95% of image creators have at least one NSFW image, and 30% exclusively creating NSFW images. The "Human attributes" theme for images (representative tags include woman, lips, head, etc.) is also significantly different from models. It is the primary focus for most image creators, with 71% of creators exclusively creating "Human attributes" images. Overall, the results confirm that deepfake and NSFW content are prevalent themes among image creators, highlighting the necessity for moderation.

## 4.3 Relationships between Themes

The prior subsection has revealed clear preferences for certain themes of models and images within the Civitai community. Abusive themes (NSFW & deepfakes) seem particularly prevalent for images. That said, deepfakes themselves are not inherently abusive; rather, they only become abusive when intertwined with other themes. For instance, a satirical deepfake portraying a politician may be innocuous, whereas one depicting a celebrity naked is problematic. Thus, we next examine the co-occurrence of themes, measured using the phi coefficient ($\phi$). Figure 2 presents the category phi correlation ($\phi$) between each pair of themes, as a heatmap.

Perhaps unsurprisingly, we find that the theme "Deepfakes" and "NSFW content" *are* positively correlated for both models ($\phi = 0.17$) and images ($\phi = 0.10$). This finding indicates a likelihood that deepfakes are associated with NSFW content on Civitai, raising concerns about NSFW deepfakes [2]. Moreover, we observe a positive correlation of "Deepfakes" and "Human attributes" (model: $\phi = 0.25$; image: $\phi = 0.18$). Further investigation reveals that such a relationship is particularly evident in the context of celebrity deepfake models. Within the subset of 5,868 models tagged with the themes "Deepfakes" and "NSFW content", 55.74% (3,271 models) are also associated with celebrity tags.[3] These results confirm that deepfakes in Civitai are likely to be abusive, given the frequent co-occurrence of deepfakes alongside NSFW content, as well as real human attributes. Moreover, themes involving real-world celebrities combined with NSFW deepfakes underscore a arguable necessity for better moderation.

## 5 EXPLORING THE CREATION OF ABUSIVE IMAGES (RQ2)

The previous section has exposed the presence of models specifically designed to generate abusive content, alongside a wealth of images

---

[3]Tags are celebrity, celeb, politician, actress, actor, singer, musician, kpop idol, idol, influencer, Instagram model, streamer, YouTuber, artist

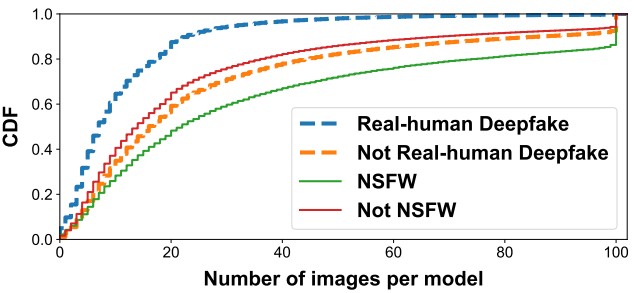

Figure 3: Comparison of the distribution of productivity among distinct types of models, measured as the number of images per model.

generated using those models. Next, we focus on the popularity of these models, and who they target.

***Popularity of deepfake and NSFW models.*** We first inspect the "popularity" of real-human deepfake and NSFW models, as this can offer moderators a useful lens into the productivity of abusive models. We measure popularity based on the number of images that have been created using those models. Recall, that the Civitai API returns whether a model is dedicated to deepfakes or NSFW. Rather than using the tags, we therefore next use these labels to classify each model. Overall, 13,516 models (15.53%) are classified as real-human deepfakes, and 7,614 models (8.75%) are classified as generating NSFW content.

Based on this classification, we see that NSFW and deepfake models are commonly used by image creators. These models have been used to produce 149,227 (5.46%) and 261,432 (9.54%) unique images, respectively. This again implies that abusive models play an important role in Civitai, where a notable portion of images are produced by these models. However, their respective importance has some nuance. Figure 3 compares the distribution of popularity of each type of model, as measured by the number of images generated per model. Compared with non-NSFW models ($mid = 14, \mu = 24.20$), we see that NSFW models are used to generate *more* images ($mid = 22, \mu = 36.36$). In contrast, real-human deepfake models are used to generate *fewer* images ($mid = 7, \mu = 11.08$) than the non-real-human deepfake models ($mid = 16, \mu = 27.88$). Even though 15.53% of models are labeled as generating real-human deepfake images, they only contribute 5.46% of all images. Thus, even though NSFW models are more popular than the average, the real-human deepfake models appear more niche, with a smaller group using them. Naturally, this does not downplay the harm that such models can cause.

***Use of NSFW prompts.*** A curious finding is that 88.40% of images reported above as NSFW are generated using non-NSFW models. We suspect this is because many models can easily be repurposed for generated NSFW content, using appropriate prompts. This dramatically increases the complexity of moderating such model sharing, motivating us to examine how much NSFW content appears in prompts.

Recall, to explore this, we use OpenAI's moderation API to derive a NSFW score (0 to 1) for each prompt based on the prompt's text.

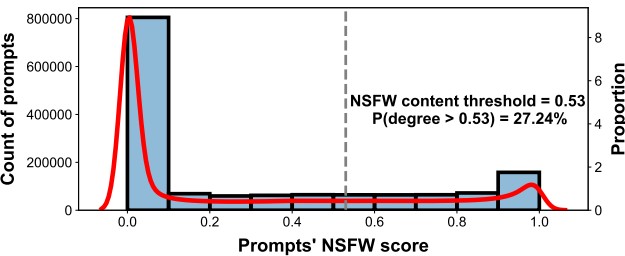

Figure 4: Distribution of prompts' NSFW score. Prompt with a degree exceeding the threshold (0.53) will be reported as NSFW prompt by OpenAI's moderation API.

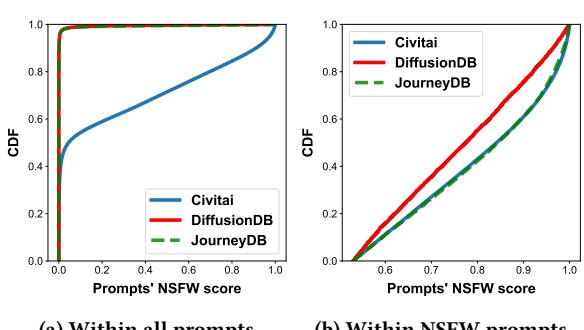

(a) Within all prompts          (b) Within NSFW prompts

Figure 5: Comparison of the distribution of prompts' NSFW score between our Civitai dataset and other two selected prompt datasets.

Figure 4 presents the distribution of the NSFW score for all prompts. Here, OpenAI's default threshold for tagging a prompt as NSFW is 0.53. We see a notable presence of NSFW prompts (score > 0.53) in the data: 404,330 (27.24%). Moreover, for those NSFW images that only employ non-NSFW models, 41.91% of them are generated with NSFW prompts. This suggests that NSFW prompts can easily repurpose models to generate NSFW images, even if these models not designed for NSFW material. Thus, there is extensive scope to repurpose non-NSFW models for NSFW purposes.

Additionally, we notice the distribution of prompts' scores appears to be bimodal with the main peak at around = 0 and a lower peak around score = 1. Notably, 39.12% of NSFW prompts contain a very high NSFW score (> 0.9). This indicates that certain creators tend to include extensive NSFW content in prompts' text. Moreover, Figure 5 illustrates the distribution of NSFW score for prompts within Civitai, Stable Diffusion Discord, and Midjourney. Recall that these prompts involve 1,534,922 prompts from Civitai, 1,528,512 prompts from *DiffusionDB* [39], and 1,466,88 prompts from *JourneyDB* [36]. This figure depicts the distribution of prompts' NSFW scores over two groups of prompts respectively – *all prompts* (a), and those reported as *NSFW prompts* (scores > 0.53) (b). Regarding *all prompts*, Civitai ($mid = 0.045, \mu = 0.286$) possesses a higher distribution of NSFW scores in the prompts, compared to both Midjourney ($mid = 0, \mu = 0.007$) and Stable Diffusion Discord ($mid = 0, \mu = 0.005$) (Figure 5a). When it comes to only NSFW

| Occupation | #Models | #Images (NSFW%) | Representatives (#Images) |
|---|---|---|---|
| **Actress** | 3,916 | 50,843 (9.73%) | Emma Watson (648), Natalie Portman (542), Ana De Armas (500), Alexandra Daddario (445), Scarlett Johansson (398) |
| **Model** | 1,663 | 19,323 (13.31%) | Emily Bloom (187), Cara Delevingne (150), Kendall Jenner (125), Jenna Ortega (110), Nicola Cavanis (100) |
| **Actor** | 717 | 7,877 (3.44%) | Henry Cavill (271), Fares Fares (135), Nicolas Cage (107), Arnold Schwarzenegger (88), Harrison Ford (82) |
| **Singer** | 757 | 7,296 (10.36%) | Billie Eilish (253), Dua Lipa (230), Taylor Swift (221), Avril Lavigne (214), Britney Spears (170) |
| **Internet influencer** | 296 | 3,323 (13.12%) | Belle Delphine (164), Brooke Monk (100), Ricardo Milos (74), Dasha Taran (71), Kris H Collins (67) |
| **Character** | 225 | 2,698 (10.08%) | Hermione Granger (99), Jill Valentine (87), Sabine Wren (69), 2B (61), El Chavo del Ocho (60) |
| **Pornstar** | 210 | 2,376 (18.56%) | Katja Kean (76), Simone Peach (60), Teagan Presley (59), Alex Coal (58), Anita Blond (50) |
| **Adult Model** | 275 | 2,239 (8.35%) | Lucid Lavender (64), Matthew Rush (39), Sean Cody (39) Hailey Leigh (36), Bunny Colby (34) |
| **Streamer** | 145 | 1,745 (15.70%) | Valkyrae/Rachell Hofstetter (166), Alexandra Botez (80), Sasha Grey (78), Andrea Botez (62) |
| **Idol** | 166 | 1,239 (7.75%) | Akina Nakamori (52), Cherprang Areekul (38), Song Yi (37), Yuino Mashu (36) Kim Ji-Woo (35) |

**Table 2: Top-10 occupations of celebrities involved in creation of deepfake models, ranked by their counts of derivative images. "#Models/#Images" presents the number of deepfake models/derivative images containing person names within corresponding occupation. "NSFW%" shows the percentage of images labeled as NSFW by Civitai API.**

prompts, prompts in Civitai ($mid = 0.841$, $\mu = 0.816$) and Midjourney ($mid = 0.844$, $\mu = 0.815$) contains noticeably higher NSFW score than those in Stable Diffusion Discord ($mid = 0.775$, $\mu = 0.771$) (Figure 5b). A one-sided two-sample Kolmogorov–Smirnov test reports that Civitai still holds a significantly ($p < 0.001$) higher distribution of NSFW score in prompts than Stable Diffusion Discord ($D = 0.150$) and Midjourney ($D = 0.023$) though. These findings indicate that these emerging AIGC social platforms may face a considerable influx of NSFW prompts, alongside a inclination among creators to use prompts to repurpose even non-NSFW models.

***Exploration of victims.*** As implied by our thematic analysis, real-world celebrities may have been directly targeted within deepfakes using models from Civitai (§4). We next inspect which celebrities and industries are the main victims in the prompts. We extract person names with their occupations from the textual usage descriptions of real-human deepfake models (see §3.2). In all, within deepfake models, ChatGPT identifies 8,297 distinct person names from 10,170 (75.24%) models, as well as 116,994 (78.40%) images generated using these models. This confirms a prevalence among image creators to target celebrities when creating real-human deepfakes.

Table 2 summarizes the top-10 occupations and statistics of the corresponding models and images. We find that celebrities from three industries are the main targets of deepfakes on Civitai: entertainment (*e.g.,* actress/actor, model, and singer), adult (*e.g.,* pornstar and adult model), and social media (Internet influencer and streamer). Interestingly, models associated with celebrities from social media industries are more common (14.01% labeled as NSFW) than targeting celebrities from entertainment (10.01%), or even

adult (13.61%) industries. Through manual inspection, we find that most of them are either closely associated with subscription platforms (*e.g.,* Belle Delphine with OnlyFans and Andrea Botez with Fanhouse) or well-known as Instagram models (*e.g.,* Kris H Collins and Brooke Monk). Whereas prior work has revealed a prominence of victims from entertainment and politics [7], only 841 (1.71%) deepfake images target politicians on Civitai. Instead, our findings highlight that it is far more common to target online celebrities.

## 6 EXPLORING USER ENGAGEMENT WITH ABUSIVE MODELS AND IMAGES (RQ3)

Prior literature has underscored the importance social engagement in encouraging users to share more abusive media [26, 29]. Civitai allows users to interact and, for example, post comments and likes on models or the images that they generate. Inspired by this, we inspect the level of social engagement received by models and images labeled as abusive.

***Metrics to quantify engagement.*** We rely on several metrics to quantify social engagement with models and images:

- *Number of downloads/views:* The total number of times that a model has been downloaded or an image has been viewed.
- *Number of favorites:* The total number of favorites that a model or image has received.[4]
- *Number of comments:* The total number of comments that a model/image has received.
- *Rating score:* The overall rating score (0 to 5) the model possesses (this is not supported for images).
- *Buzz:* The volume of Buzz a model/image accumulates by receiving tips from users. Here, the Buzz is the in-site digital currency on Civitai [8].

***Engagement with deepfake and NSFW models and images.*** Using the above metrics, we inspect whether abusive content triggers more active user engagement. For this, we first group models and images as either real-human deepfakes or NSFW, independently. We then perform the Mann-Whitney U test to assess in-group difference on each of the aforementioned metrics.

Table 3 summarizes the comparison results on the user engagement for the model/image groups, categorized by their label as real-human deepfakes or NSFW. All comparisons possess statistical significance ($p < 0.001$), which suggests that models and images related to abusive content receive different engagement levels. We find that *NSFW models and images* have a *higher* volume across almost all engagement metrics, on average. Compared with non-NSFW ones, NSFW models and images are more likely to be downloaded or viewed (models: 3.32x; images: 1.18x), gain more favorites (models: 3.22x; images: 1.63x) and more tips (models: 1.92x; images: 1.53x). Additionally, NSFW models not only attain a higher rating score (1.96x non-NSFW models' rating score), they also trigger viewers to leave more comments and tips (1.09x non-NSFW models' comments volume).

Interestingly, these trends get reversed for **real-human deefakes**. Real-human deepfake models or images get a *lower* volume of all engagement metrics on average. For example, compared with not

---

[4]While a model has a favourites count, an image's favorites are represented by two emoji-based reactions, "like" and "heart", left by viewers under the image.

|  | Real-human deepfake | | NSFW | |
|---|---|---|---|---|
|  | Mean diff (True *vs.* False) | p-value | Mean diff (True *vs.* False) | p-value |
| Number of downloads | 582.42 < 1539.32 | *** | 3842.83 > 1155.68 | *** |
| Number of favorites | 66.55 < 237.61 | *** | 569.23 > 176.72 | *** |
| Number of comments | 2.32 < 3.24 | *** | 4.53 > 2.31 | *** |
| Rating score | 3.27 < 3.43 | *** | 3.58 > 3.28 | *** |
| Buzz | 6.64 < 45.24 | *** | 70.33 > 36.54 | *** |

**(a) Engagement with models**

|  | Real-human deepfake | | NSFW | |
|---|---|---|---|---|
|  | Mean diff (True *vs.* False) | p-value | Mean diff (True *vs.* False) | p-value |
| Number of views | 747.25 < 958.80 | *** | 1030.18 > 866.55 | *** |
| Number of favorites | 1.51 < 2.54 | *** | 3.09 > 1.89 | *** |
| Number of comments | 0.017 < 0.032 | *** | 0.029 < 0.033 | *** |
| Rating score | - | - | - | - |
| Buzz | 0.15 < 0.47 | *** | 0.55 > 0.36 | *** |

**(b) Engagement with images**

**Table 3: Comparison on metrics of user engagement by the Mann-Whitney U test between models/images groups categorized by their label as real-human deepfakes or NSFW. "Mean diff" column shows the comparison results of the mean value of corresponding metrics between two groups; ***: $p < 0.001$.**

real-human deepfake ones, real-human deepfake models and images are less likely to be downloaded or viewed (models: 0.38x; images: 0.78x), gain less favorites (models: 0.28x; images: 0.59x) and less tips (models: 0.15x; images: 0.32x).

***Engagement and model productivity.*** One explanation for the above is that NSFW models gain more engagement by being used to produce more images than deepfake ones. We refer to this as a model's "productivity". To explore this relation, we calculate the Pearson correlation between each models' volume of generated images vs. the users' engagement with these models. Table 4 presents the results of the correlation analysis across the models, grouped as real-human deepfakes or NSFW. Confirming our intuition, all engagement metrics exhibit a significant ($p < 0.001$) positive correlation with models' productivity ($r > 0$). In other words, the models that are used to generate images more often tend to also gain more social engagement. Moreover, given that NSFW models ($mid = 22, \mu = 36.36$) have produced more images than deepfake models ($mid = 16, \mu = 27.88$), this correlation can explain why NSFW models trigger more user engagement than deepfake ones. The higher productivity and engagement for these models suggests that NSFW content existing within a wider community of active users.

## 7 EXPLORING THE NETWORK POSITIONS OF ABUSIVE CREATORS (RQ4)

One potential explanation for the above prevalence of abusive models and images is that help creators to gain a greater social status. To explore this, we next investigate whether the sharing of abusive models and images is correlated with a creator's social network position (*e.g.,* centrality).

|  | Real-human deepfake | | NSFW | |
|---|---|---|---|---|
|  | Pearson's $r$ | p-value | Pearson's $r$ | p-value |
| Number of downloads | 0.344 | *** | 0.287 | *** |
| Number of favorites | 0.313 | *** | 0.349 | *** |
| Number of comments | 0.288 | *** | 0.223 | *** |
| Rating score | 0.281 | *** | 0.408 | *** |
| Buzz | 0.032 | *** | 0.113 | *** |

**Table 4: Correlation analysis by measuring Pearson's $r$ between models' volume of generated images and users' engagement with the models. ***: $p < 0.001$.**

### 7.1 Social Network Definition

We induce a follower network based on users' following connections. If a user (*followee*) is followed by another user (*follower*), we assign a directed link from the follower to the followee. The resulting social network is a directed graph consisting of 214,218 nodes and 1,731,805 edges. Among these users, we focus on *active creators* who have shared at least 3 models or images. We further categorize these creators as NSFW/deepfake if they have contributed at least 3 NSFW/deepfake models or images. In all, there are 26,180 (12.22%) active creators who generate 1,699,606 (98.14%) connections on the follower network. 19,160 (73.18%) are NSFW creators and 2,591 (9.89%) are deepfake creators. This means that the follower network is predominately led by a small group of active creators, who commonly share models and images. This unsurprisingly seems to play a key role in forming the connections.

### 7.2 Centrality Analysis

Graph centrality is a metric to assess users' positions on a social network [5]. We calculate three centrality metrics to quantify the creators' network positions [30, 47]: (*i*) *Betweenness:* Creators with higher betweenness centrality hold a brokerage position, connecting different communities within the social network; (*ii*) *Indegree:* Creators with higher indegree are more popular, with more followers; (*iii*) *PageRank:* Creators with higher PageRank are followed by other users, who have high influence.

To inspect whether abusive creators have different network positions, we calculate the above centrality metrics for all users. Figure 6 presents the results for the three centrality features between the different types of creators. We observe that abusive creators *do* possess higher centrality within the follower network.

Model creators who share deepfake models hold higher values for all the three centrality metrics, compared to those who never share any abusive models (`Neither < Both, Deepfake-only`). Similarly, image creators who share NSFW images also hold higher values for all the three centrality features (`Neither, Deepfake-only < Both, NSFW-only`). Thus, abusive creators do demonstrate a higher level of centrality within the follower network, indicating their distinct position as brokerage position, popular followees, or neighbors of influencers. We posit that this also impacts the earlier engagement metrics, as users with more central positions in the social graph will attain higher exposure.

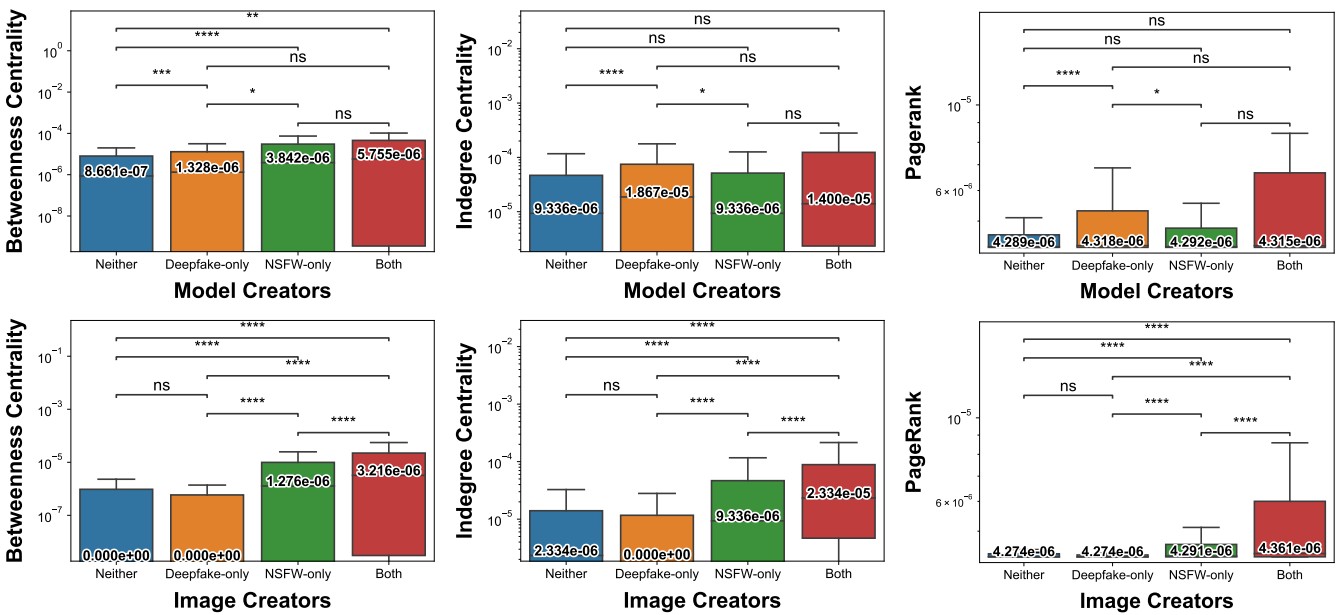

**Figure 6: Comparison of network centrality between diverse types of creators. "Both" refers to the creators categorized as both NSFW and deepfake at the same time. The statistical significance is reported by the Kruskal-Wallis test with Dunn's post-hoc test;** ****: $p < 0.0001$, ***: $p < 0.001$, **: $p < 0.01$, *: $p < 0.05$, **ns: none significance.**

## 8 RELATED WORK

***Platforms for AI models.*** Previous studies have looked at online platforms for AI models, with a particular emphasis on traditional platforms like GitHub and Huggingface. These investigations cover a wide range of perspectives, including machine learning [27, 37], software engineering [20, 37], and social computing [1, 41]. Additionally, there are also studies that put forward innovative designs for these platforms [18, 21]. In contrast, Civitai and other AIGC social platforms also serve as a hub to showcase AIGC, and an online community for AI creators, attracting a diverse user base that extends beyond programmers and computer scientists. To the best of our knowledge, this is the first large-scale empirical study of an emerging AIGC social platform.

***Abuse of generative AI.*** Several studies have examined the abuse of generative AI. There are two perspectives closely related to our work. The first issue concerns the spread of misinformation through deepfakes [43]. Multiple studies have looked into the prevalence of deepfakes on social media and their potential impact on security and safety [7, 15, 24, 28, 32, 45]. The second issue involves the creation of NSFW content more generally. Numerous studies have highlighted the significant increase in AI-generated NSFW content on the Internet, particularly on social media platforms. Concerns have been raised about the lack of regulation and moderation of this content, and the potential impact it may have on the online environment and community building [6, 13, 17, 38, 41]. In contrast, Civitai and other AIGC social platforms offer more than just AI-generated images — they include generative AI models that produce the abusive images. Overall, our research complements prior studies by providing insights not only from the image angle, but also from

the model and creator perspective. We argue this can help in better regulating and moderating potentially abusive models and images.

## 9 CONCLUSION AND DISCUSSION

***Summary and implications.*** This paper performed a large-scale study of the creative ecosystem of Civitai. Our analysis reveals abusive use of generative models, mainly revolving around NSFW content and deepfakes. Moreover, we flag several crucial points related to the use of NSFW prompts, emerging deepfake attacks on social media celebrities, and users' active engagement with abusive models and images. This motivates the need to better develop moderation tools by analysing creators' network positions.

***Limitations and future work.*** Our research is based solely on Civitai. Moving forward, we hope to include additional AIGC platforms such as PixAI and Tensor.art. This expansion will allow us to gain a wider perspective on the patterns of AI-generated abuse across various platforms. Furthermore, so far, our focus is limited to the exploration of NSFW content and deepfake abuses through generative models. We wish to explore other forms of misuse, such as copyright infringement, the creation of offensive memes, and the production of false information.

***Ethics consideration.*** Our study is based on public data from the Civitai RESTful API. We do not attempt to de-anonymise users. We only use the data to understand users' behaviors associated with abusive AIGC and discuss moderation strategies for Civitai. Our analyses follow Civitai's data policies.

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
