# OpenReview forum: "Exploring the Use of Abusive Generative AI Models on Civitai"
_acmmm.org/ACMMM/2024/Conference — MM2024 Poster_

### Official Review · Reviewer_ngPi · 2024-05-25

**Rating:** 4
**Confidence:** 2

**Summary:**

The authors conducted a comprehensive study of the AI-generated content (AIGC) on Civitai. Specifically, they studied the types of AI models and generated imageries that are generated and spread within the Civitai’s community. Their analysis shows that many models and images are directed towards specific themes (i.e., NSFW and deepfake content), which can infringe on one’s privacy and copyright. This analysis highlights the abuse of AIGC in the broader community. Overall, this paper is a nice analysis and (possibly) resource paper. However, I am unsure whether this kind of content is suitable for the ACM Multimedia Conference.

**Strengths:**

- [Research Topic] With the proliferation of AIGC, this research paper provides a breakdown of the popular AIGC content that is viral and spreading over the Internet. The analysis also highlights the common abuse of AIGC, such as generating NSFW or deepfake content.
- [Data Scale] The authors collected a large amount of AIGC data (87k+ models’ metadata and 2.7mil+ images). The large amount of data helps ensure the credibility of their findings.
- [Detailed Analysis] The authors conducted a series of comprehensive analyses, from models to images to prompts. These analyses provide a comprehensive overview of the current state of AIGC on Civitai. Additionally, the authors highlight that it is prevalent for end-users to use NSFW prompts on non-NSFW models.

**Limitations:**

- Resource/Analysis Oriented] While the research topic is exciting, the paper is more oriented as a resource and/or analysis paper. It would be more exciting if the authors could propose or devise some methodology to implement safeguards for non-NSFW models, preventing them from generating NSFW content.
- [Future Works/Directions] While this analysis of AIGC content on Civitai highlights the importance of preventing misuse in the broader community, it is unclear what this resource can also be used for. Nevertheless, this limitation doesn't pose a strong factor for rejection.

**Suitability:**

3

---

### Official Review · Reviewer_RZtz · 2024-05-26

**Rating:** 3
**Confidence:** 3

**Summary:**

This study performs an empirical study of an AIGC social platform to examine the use for generating abusive content. A dataset of 87K models and 2M images is composed.

**Strengths:**

(1) The research topic is important and interesting.

(2) The analysis is comprehensive.

**Limitations:**

(1) The findings are based on one platform. As a result, they may not be generalizable.

(2) It would be better to include a more comprehensive discussion about the implications. What are the insights for policies? The current discussion may lack in-depth thoughts.

Minor:

(1) The manuscript should be better structured. There is still much white space (e.g., before Section 7.1)

**Suitability:**

3

---

### Official Review · Reviewer_RfaL · 2024-05-27

**Rating:** 6
**Confidence:** 3

**Summary:**

The paper presents a comprehensive empirical study of the largest AI-generated content (AIGC) social platform, Civitai. It compiles and analyzes a dataset of 87,042 models and 2,740,149 images, revealing significant issues with the generation and dissemination of NSFW content and deepfakes. The study identifies the prevalence of abusive content, the engagement patterns of users with such content, and the social network positions of creators who produce abusive material, highlighting the need for improved moderation strategies on AIGC platforms.

The authors provided interesting insights into the disturbing prevalence and engagement of NSFW content and deepfakes. Their analytical methodology, which includes robust statistical and social network analyses, uncovers the significant impact of abusive content. Although the paper does not introduce novel theoretical models or algorithms and is limited to the Civitai platform, the originality of the dataset and the scale of the empirical study are noteworthy. The findings reveal critical issues and underscore the urgent need for improved content moderation strategies. This paper is a crucial contribution to the field, offering valuable practical implications that can guide future efforts to combat abusive AI-generated content.

**Strengths:**

1. First Large-Scale Study of Civitai: The paper offers the first large-scale empirical analysis of an AIGC social platform, specifically focusing on Civitai. This novelty adds significant value to the research, providing new insights into the dynamics of AI-generated content on this platform.
2. Comprehensive Dataset: The paper constructs a large and comprehensive dataset from the Civitai platform, including metadata for 87,042 models and 2,740,149 images. This extensive dataset provides a robust foundation for empirical analysis and enhances the credibility of the study's findings.
3. Detail Analysis: The study thoroughly examines the prevalence and nature of NSFW content and deepfakes, including their correlation with user engagement metrics and social network positions. This detailed analysis sheds light on critical issues related to abusive content, contributing to the understanding and potential mitigation of such problems on AIGC platforms. The paper employs various analytical techniques, including statistical methods, social network analysis, and automated annotation tools, which enhances the robustness of the analysis and provides a comprehensive understanding of the data.

**Limitations:**

1. Potential Bias in Annotation: The paper uses ChatGPT for theme extraction and OpenAI’s moderation API for NSFW content identification. These tools, while powerful, may have biases and limitations in accurately classifying content. The reliance on these tools without extensive validation might introduce biases in the dataset annotation, potentially affecting the accuracy of the findings.The paper uses ChatGPT for theme extraction and OpenAI’s moderation API for NSFW content identification. These tools, while powerful, may have biases and limitations in accurately classifying content. Impact: The reliance on these tools without extensive validation might introduce biases in the dataset annotation, potentially affecting the accuracy of the findings.
2. Over-reliance on Engagement Metrics: The study heavily relies on engagement metrics such as downloads, views, favorites, and comments to draw conclusions about the popularity and impact of abusive content. While these metrics are useful, they may not fully capture the nuanced user interactions and motivations behind engaging with abusive content. Additional qualitative analysis could provide a more comprehensive understanding.

**Suitability:**

3

---

### Meta-Review · Area_Chair_DFFM · 2024-07-04

**Recommendation:** Accept (Poster)
**Confidence:** 5

**Metareview:**

This paper presents a very detailed empirical study of Civitai, a large AI-generated content social platform. Specifically, the authors have organized and analyzed a dataset of 87,042 models and 2,740,149 images, revealing significant issues with the generation and dissemination of deepfakes and not safe content. All the reviewers agree that the topic is very interesting, the conducted analysis is very detailed, and the dataset organized of big relevance for several research communities. Some concerns are raised with the relevance for ACM-MM (maybe other better fits in terms of conferences could be ICWSM or CSCW) and the fact that the work is mainly a dataset curation and data analyses and not an algorithmic contribution. In any case, there is consensus on having this work at ACM-MM.